# Evidence for Anti-Inflammatory Activity of Isoliquiritigenin, 18β Glycyrrhetinic Acid, Ursolic Acid, and the Traditional Chinese Medicine Plants *Glycyrrhiza glabra* and *Eriobotrya japonica*, at the Molecular Level

**DOI:** 10.3390/medicines6020055

**Published:** 2019-05-10

**Authors:** Jun-Xian Zhou, Michael Wink

**Affiliations:** Institute of Pharmacy and Molecular Biotechnology, Heidelberg University, Im Neuenheimer Feld 364, Heidelberg 69120, Germany; junxian.zhou@stud.uni-heidelberg.de

**Keywords:** traditional Chinese medicine (TCM), plant secondary metabolites (PSMs), NF-κB, ICAM-1, TNF-α, iNOS, COX-2, nuclear translocation

## Abstract

**Background:** We investigated the effect of root extracts from the traditional Chinese medicine (TCM) plants *Glycyrrhiza glabra* L., *Paeonia lactiflora* Pall., and the leaf extract of *Eriobotrya japonica* (Thunb.) Lindl., and their six major secondary metabolites, glycyrrhizic acid, 18β glycyrrhetinic acid, liquiritigenin, isoliquiritigenin, paeoniflorin, and ursolic acid, on lipopolysaccharide (LPS)-induced NF-κB expression and NF-κB-regulated pro-inflammatory factors in murine macrophage RAW 264.7 cells. **Methods:** The cytotoxicity of the substances was determined using the 3-(4,5-dimethylthiazol-2-yl)-2,5-diphenyltetrazolium bromide (MTT) method. RAW 264.7 cells were treated with LPS (1 μg/mL) or LPS plus single substances; the gene expression levels of NF-κB subunits (RelA, RelB, c-Rel, NF-κB1, and NF-κB2), and of ICAM-1, TNF-α, iNOS, and COX-2 were measured employing real-time PCR; nitric oxide (NO) production by the cells was quantified with the Griess assay; nuclear translocation of NF-κB was visualized by immunofluorescence microscopy with NF-κB (p65) staining. **Results:** All the substances showed moderate cytotoxicity against RAW 264.7 cells except paeoniflorin with an IC_50_ above 1000 μM. *Glycyrrhiza glabra* extract and *Eriobotrya japonica* extract, as well as 18β glycyrrhetinic acid and isoliquiritigenin at low concentrations, inhibited NO production in a dose-dependent manner. LPS upregulated gene expressions of NF-κB subunits and of ICAM-1, TNF-α, iNOS, and COX-2 within 8 h, which could be decreased by 18β glycyrrhetinic acid, isoliquiritigenin and ursolic acid similarly to the anti-inflammatory drug dexamethasone. NF-κB translocation from cytoplasm to nucleus was observed after LPS stimulation for 2 h and was attenuated by extracts of *Glycyrrhiza glabra* and *Eriobotrya japonica*, as well as by 18β glycyrrhetinic acid, isoliquiritigenin, and ursolic acid. **Conclusions:** 18β glycyrrhetinic acid, isoliquiritigenin, and ursolic acid inhibited the gene expressions of ICAM-1, TNF-α, COX-2, and iNOS, partly through inhibiting NF-κB expression and attenuating NF-κB nuclear translocation. These substances showed anti-inflammatory activity. Further studies are needed to elucidate the exact mechanisms and to assess their usefulness in therapy.

## 1. Introduction

The nuclear transcription factor NF-κB plays a key role in inflammatory response. NF-κB is composed of homo- or heterodimers of Rel protein family members (RelA (p65), RelB, Rel, NF-κB1 (p105/p50), and NF-κB2 (p100/p52)) [1]. In the cytoplasm, NF-κB is complexed with its inhibitory protein IκB. Upon various extracellular signals, such as lipopolysaccharide (LPS) from Gram-negative bacteria, stress, cytokines (TNF-α, IL-1), and others [2,3], IκB is phosphorylated by activated IκB kinase (IKK) and ubiquitinated. Then, the free NF-κB moves to the nucleus where it binds to various promotors and regulates the expression of corresponding genes [2,4], such as cytokines (TNF-α, IL-6, etc.), cell adhesion molecules (ICAM-1, VCAM-1, etc.), transcription factors (c-Rel, IκBα, p105, etc.), stress response genes (angiotensin II, COX-2), and enzymes (collagenase 1, iNOS), and so on [3,5]. These proteins play important roles in inflammatory response, cell survival, proliferation and adhesion, cancer, and the immune system.

Dysregulated NF-κB is an important player in the pathophysiology and progression of several diseases [6]. Tumor necrosis factor-α (TNF-α), mainly produced in macrophages through LPS-mediated NF-κB activation [7,8], is associated with various inflammatory conditions such as sepsis, rheumatoid arthritis, inflammatory bowel disease, and so on [9,10]. Cyclooxygenase is a key catalytic enzyme in arachidonic acid metabolism to prostanoids (thromboxanes, prostacyclins and prostaglandins) [11]. NF-κB activation by inflammatory stimuli such as bacterial LPS, cytokines (TNF-α, IL-1, etc.), or others can cause a rapid increase in COX-2 expression [12,13,14,15]. The increased COXs and prostaglandins may promote the development and progression of some diseases such as cancer, pain, inflammation, Alzheimer’s disease, and others [16,17]. The cell adhesion molecule family member ICAM-1 [18] is widely expressed in various cells and undergoes fast activation and increase by pro-inflammatory cytokines such as interleukin-1, TNF-α, interferon-γ, or LPS, via NF-κB [18,19,20]. Elevated ICAM-1 is associated with inflammatory syndromes, inflammatory conditions, cancer, and vascular disease development [20,21,22]. Inducible nitric oxide synthase (iNOS) is the main enzyme that produces nitric oxide (NO) from L-arginine under physiological conditions [23]. iNOS is usually activated by NF-κB and signal transducer and activator of transcription 1 (STAT-1α) under immunological or inflammatory challenge [24,25]. The free radical NO plays important roles in the cardiovascular, pulmonary, nervous, and immune systems [26,27,28]. Over-production of NO can cause toxic effects, tissue damage, and other pathological changes [29,30].

The non-steroidal anti-inflammatory drugs (NSAIDs) salicylic acid and aspirin inhibit cyclooxygenases. They also modulate NF-κB activation [14,31]. However, NSAIDs may cause adverse effects in the gastrointestinal tract and in the kidneys [32], and several selective developed COX-2 inhibitors showed some cardiovascular risks [33].

Many plant secondary metabolites (PSMs) exhibit anti-inflammatory activities [34], which can explain the use of many medicinal plants to treat pain and inflammation [35,36]. The anti-inflammatory activities of three extracts from traditional Chinese medicine (TCM) plants (the root extracts of *Glycyrrhiza glabra* (Ge), *Paeonia lactiflora* (mixed with *Paeonia veitchii*) (Pe), and the leaf extract of *Eriobotrya japonica* (Ue)) and six of their major secondary metabolites (the triterpenes glycyrrhizic acid (ga) and 18β glycyrrhetinic acid (18ga), the flavonoids liquiritigenin (liq) and isoliquiritigenin (iso) from *G. glabra*, monoterpene paeoniflorin (pae) from *P. lactiflora*, and triterpenoid ursolic acid (urs) from *E. japonica*) have been studied. The plant extracts and the structures of the PSMs were previously introduced [37]. We measured the cytotoxicity of the nine substances (plant extracts and PSMs) to illustrate the safe use of these substances as potential drugs. We measured their effects on LPS-induced inflammatory responses including nitric oxide production, gene expressions of NF-κB subunits and pro-inflammatory factors TNF-α, ICAM-1, COX-2, iNOS, and NF-κB and nuclear translocation in RAW 264.7 cells, in order to elucidate the molecular mechanisms underlying their anti-inflammatory activities. These results help to evaluate these substances to be developed into anti-inflammatory drugs.

## 2. Materials and Methods

### 2.1. Plant Materials and Chemicals

The origins of the plant materials (*Glycyrrhiza glabra*, *Paeonia lactiflora*, and *Eriobotrya japonica*) and their extraction processes were previously described [37]. Lipopolysaccharide (LPS) from *Escherichia coli* O55:B5, 3-(4,5-dimethylthiazol-2-yl)-2,5-diphenyltetrazolium bromide (MTT), doxorubicin, sodium nitrite, sulfanilamide, N-1-napthylethylenediamine dihydrochloride (NED), and dexamethasone were obtained from Sigma-Aldrich (Darmstadt, Germany). NF-κB p65 polyclonal antibody and Goat anti-Rabbit IgG (H+L) highly cross-adsorbed secondary antibody, Alexa Fluor 488, were purchased from ThermoFisher Scientific (Karlsruhe, Germany). Glycyrrhizic acid, 18β glycyrrhetinic acid, isoliquiritigenin, liquiritigenin, paeoniflorin, and ursolic acid came from Baoji Herbest Bio-Tech (Baoji, China).

### 2.2. Cell Culture and Cell Viability Assay with MTT

The murine macrophage cell line RAW 264.7 was a gift from PD Dr. Volker Lohmann (Department of Infectious Diseases, Molecular Virology, Centre for Integrative Infectious Disease Research (CIID), Heidelberg University, Heidelberg, Germany). The cell culture and MTT method were the same as in the previous publication [38].

### 2.3. Griess Assay

Nitric oxide has a very short half-life and in seconds it can be oxidized into other radicals such as nitrite (NO_2_^−^), nitrate (NO_3_^−^), nitrogen dioxide (NO_2_), dinitrogen trioxide (N_2_O_3_), and peroxynitrite (ONOO^−^), etc., in vivo. The total production of these radicals is referred to as nitric oxide (NO) and can be evaluated by measuring the stable end products nitrite and nitrate (nitrites) by a Griess assay [39,40]. The Griess assay was carried out following the protocol of Griess Reagent System (Promega). 1% sulfanilamide solution was prepared in 5% phosphoric acid, and 0.1% N-(1-naphthyl) ethylenediamine dihydrochloride (NED) solution in water. All the solutions were protected from light and stored at 4 °C. Before use, the sulfanilamide solution and NED solution were allowed to equilibrate to room temperature (ca. 15 min).

Before the substance testing, a pre-experiment was performed: cells in six T25 flasks were treated with different concentrations of LPS (0.01 μg/mL, 0.025 μg/mL, 0.05 μg/mL, 0.1 μg/mL, 0.5 μg/mL, and 1 μg/mL) respectively and the NO production in each flask was analyzed at 1 h, 2 h, 3 h, 4 h, 5 h, 6 h, 7 h, 8 h, 9 h, 10 h, 11 h, 12 h, 22 h and 24 h with Griess reagents to obtain the curve of nitrite amount vs. time and the optimal dose of LPS.

RAW 264.7 cells in T25 flasks were treated with 1 μg/mL (this concentration value was set based on the pre-experiment) LPS or LPS plus various non-toxic doses of single substances for 24 h. A concentration gradient of sodium nitrite (100, 50, 25, 12.5, 6.25, 3.13 and 1.56 μM in 100 µL media) was prepared in a 96-well plate to produce a nitrite standard curve. 100 µL of the supernatant taken from the cell flasks was added into the 96-well plate in triplicate. Then, 50 µL of the sulfanilamide solution was added to all experimental samples, including the nitrite standard series, and was left to react for 5–10 min in darkness at room temperature. Afterwards, 50 µL of the NED solution was dispensed to all the samples and incubated for 5–10 min in darkness at room temperature. The absorption was measured at 550 nm with Tecan Nano Quant infinite M200 PRO Plate Reader (Tecan, Männedorf, Switzerland). The nitric oxide synthase inhibitor L-N^G^-nitroarginine methyl ester (L-NAME) [41,42,43] was used as a positive control.

### 2.4. NF-κB-Related Gene Expression Analysis by Real-Time PCR

RAW 264.7 cells were treated with 1 μg/mL LPS for 0 h, 1 h, 2 h, 3 h, 4 h, 6 h, and 8 h. The gene expressions of NF-κB1, NF-κB2, RelA, RelB, Rel, ICAM-1, TNF-α, iNOS, and COX-2 were measured by real-time PCR. According to the expression results, the time points for the expression peaks of various genes were chosen in order to test whether the substances could inhibit the gene expressions. Namely, substances were added along with LPS into the cell media. Dexamethasone, which inhibited NF-κB and TNF-α transcription, and ICAM-1 expression via NF-κB [7,44], was used as a positive control. The protocol for real-time PCR with SYBR Green I dye followed the previous publication [37]. Primers are provided in Table 1.

### 2.5. Imaging of NF-κB Nuclear Translocation

RAW 264.7 cells were grown on round glass coverslips (Thermo Scientific, Braunschweig, Germany) in a 6-well plate. They were treated with LPS (1 μg/mL) or LPS plus a single drug (30 μg/mL Ge, 10 μg/mL Pe, 30 μg/mL Ue, 10 μM ga, 20 μM 18ga, 50 μM liq, 4 μM iso, 20 μM pae and 10 μM urs) for 2 h. Dexamethasone (10 μM) was used as a positive control. Then, the cells were washed twice with cold PBS, and fixed with 4% paraformaldehyde for 15–20 min, followed by cold PBS-washing. Then, the cells were blocked with 4% skimmed milk dissolved in PBS for 1 h. After PBS-washing, cells were incubated with NF-κB (p65) polyclonal antibody (1:400) (ThermoFisher Scientific, Fisher Scientific GmbH, Schwerte, Germany) overnight at 4 °C. Followed by a 1 h incubation with Goat anti-Rabbit IgG (H+L) highly cross-adsorbed secondary antibody (1:1000) (ThermoFisher Scientific, Fisher Scientific GmbH, Schwerte, Germany) at room temperature. Afterwards, DAPI was used to stain the cell nucleus at room temperature for 5 min. Then, the coverslips were visualized by fluorescence microscopy (BZ-9000, Keyence, Keyence Deutschland GmbH, Neu-Isenburg, Germany).

### 2.6. Statistical Analysis

Results were expressed as mean ± SD from at least three independent experiments. Data analysis was performed with SigmaPlot^®^ 11.0 (Systat Software, San Jose, CA, USA) for the cytotoxicity assay and GraphPad Prism 6 (Graphpad Software, San Diego, CA, USA) for the gene expression experiments. Statistical significance was evaluated employing one-way ANOVA and when *p* < 0.05, the difference was regarded as significant (* *p* < 0.05; ** *p* < 0.01; *** *p* < 0.001; **** *p* < 0.0001).

## 3. Results

### 3.1. Cytotoxicity of Individual Substances in RAW 264.7 Cells

The cytotoxicity of the nine substances in RAW 264.7 cells at 24 h was examined using an MTT assay. As Table 2 shows, doxorubicin was very toxic, pae was not toxic with an IC_50_ value above 1000 μM, and the other substances showed moderate toxicity with IC_50_ values between 20 and 263 μM (or μg/mL).

### 3.2. Nitrite Level after LPS Stimulation in RAW 264.7 Cells 

RAW 264.7 cells were treated with different concentrations of LPS (1 μg/mL, 0.5 μg/mL, 0.1 μg/mL, 0.05 μg/mL, 0.025 μg/mL, and 0.01 μg/mL) and the nitrite level was determined at 1 h, 2 h, 3 h, 4 h, 5 h, 6 h, 7 h, 8 h, 9 h, 10 h, 11 h, 12 h, 22 h and 24 h. Nitrite production was time-dependent and LPS dose-dependent (Figure 1). 1 μg/mL LPS induced the highest concentration of nitrite, and the nitrite level peaked at 22 h and decreased afterwards. The cells didn’t look good and most died at 24 h. For the other experiments, 1 μg/mL LPS was used as a stimulant and the nitrite level was determined after 24 h treatment.

### 3.3. Effect of Single Substances on Nitrite Production in RAW 264.7 Cells

Three non-toxic concentrations of single substances (below IC_10_) were chosen to be added along with LPS (1 μg/mL) to RAW 264.7 cells in T25 flasks for 24 h. Cells without treatment were used as the control group. The nitric oxide synthase inhibitor L-NAME was used as a positive control. As shown in Figure 2, 1 μg/mL LPS induced a 15- to 30-fold increase in the nitrite amount, and L-NAME, Ge, Ue, 18ga, and iso inhibited nitrite production in a dose-dependent manner.

### 3.4. Effect of Single Substances on the Gene Expressions of NF-κB Subunits and Pro-Inflammatory Factors in RAW 264.7 Cells

RAW 264.7 cells were treated with LPS (1 μg/mL) or LPS plus a single substance for 1, 2, 3, 4, 6, and 8 h. Gene expressions of NF-κB subunits NF-κB1, NF-κB2, RelA, RelB, and Rel, and NF-κB-regulated genes ICAM-1, TNF-α, iNOS, and COX-2, were analyzed by real-time PCR. Non-treated cells were set as the control group (expression was 1.00). As shown in Figure 3a, LPS induced an expression increase in all the genes within 8 h. RelA (p65), RelB, and Rel peaked at 2 h, TNF-α and iNOS peaked at 3 h, NF-κB1 (p50), NF-κB2, ICAM-1, and COX-2 peaked at 4 h.

According to the NO inhibition results, the PSMs 18ga and iso were selected to test their effect on the gene expressions of the NF-κB subunits and pro-inflammatory factors. Besides, urs inhibited NO production at IC_10_ (data not shown), so urs (IC_10_ = 10 μM) was also selected. A non-cytotoxic concentration of the single substances 18ga (20 μM), iso (4 μM), and urs (10 μM), was added along with LPS to RAW 264.7 cells for 2 h to test RelA, RelB, Rel, and NF-κB2 expressions; for 3 h to test ICAM-1, TNF-α, and iNOS expressions; for 4 h to test NF-κB1; and 6 h to test COX-2 expressions (Figure 3b). Dexamethasone (10 μM) was used as a positive control in each case. Dexamethasone and the three PSMs all inhibited the tested gene expressions to some degree.

### 3.5. Effect of Single Substances on the Nuclear Translocation of NF-κB

The same non-toxic concentrations of 18ga, iso, and urs (20 μM 18ga, 4 μM iso, and 10 μM urs) as those in the NF-κB-related gene experiment, and other substances (30 μg/mL Ge, 10 μg/mL Pe, 30 μg/mL Ue, 10 μM ga, 50 μM liq, 20 μM pae), were selected to examine their effect on the nuclear translocation of NF-κB. In the inactive state, NF-κB is bound to the inhibitory protein IκB and stays in the cytoplasm. When activated by LPS, NF-κB dissociates from IκB and translocates to the nucleus. As seen in Figure 4, NF-κB was located in the cytoplasm in the control group, and after activation by LPS, NF-κB translocated to the nucleus. The PSMs 18ga, iso, and urs and plant extracts Ge and Ue could inhibit NF-κB nuclear translocation.

## 4. Discussion

Macrophages are one of the first cells in the host innate immune system to recognize and detect infectious sources such as LPS [45] and major producers of pro-inflammatory cytokines [46].

A number of medicinal plants have been used or are still in use to treat inflammations [34,35]. Some studies have reported that *Eriobotrya japonica* extract, ursolic acid, glycyrrhizic acid, 18β glycyrrhetinic acid, isoliquiritigenin, and paeoniflorin exhibited anti-inflammatory effects, partly through NF-κB and other pathways [47,48,49,50,51,52,53], but few studies have reported this for *G. glabra* extract, *P. lactiflora* extract, and liquiritigenin. Few studies have revealed the time-course changes of LPS-induced pro-inflammatory factors in detail. In Figure 1, we can see clearly how nitrite level changed during 24 h and the influence of different LPS concentrations. Below 0.1 μg/mL, the effect of LPS was not apparent. Before 8 h, the nitrite level hardly changed, but the expression of NF-κB subunits and down-stream genes (e.g., iNOS) was already activated and reached peaks within 8 h. This confirms the regulating role of NF-κB and iNOS on NO production.

In the time-course study of NF-κB subunits and NF-κB-regulated gene expressions, all the genes except COX-2 showed a bell-like curve. They reached their expression peaks within 8 h (Figure 3a). The expression of NF-κB1 (p50), NF-κB2, RelA (p65), and Rel increased up to four-fold at 2 h, NF-κB1 (p50) peaked at 4 h with a near 10-fold increase. The increases in NF-κB-regulated gene (ICAM-1, TNF-α, iNOS, and COX-2) expressions were apparent at 3 h. When we started our experiments, we stimulated the cells for 12 and 24 h with LPS, however, most of the cells were dead after that time, which might be due to the high sensitivity of the cell line. As a consequence, we selected 2 h and 4 h for NF-κB1 to test the effect of PSMs on NF-κB subunit expressions, and 3 and 6 h to test the effect of PSMs on ICAM-1, TNF-α, iNOS, and COX-2 expressions. As expected, the three PSMs 18ga, iso, and urs inhibited all the genes in a similar way to the positive control dexamethasone (Figure 3b), confirming the anti-inflammatory effect of these substances.

The positive control L-NAME inhibited NO production, but did not always have a significant inhibition effect in all experiments, which might be due to experimental conditions. Similarly, except in the gene expression of ICAM-1 and COX-2 groups, the positive control dexamethasone did have effects.

Isoliquiritigenin and ursolic acid blocked NF-κB translocation (Figure 4), which agrees with some reports that these PSMs can disturb NF-κB translocation. In other cell lines, these PSMs were involved with the inhibition of IκBα degradation NF-κB (p65) phosphorylation, as well as ERK and JNK pathways [47,54,55]. 18β glycyrrhetinic acid was reported to exert an anti-inflammatory effect by inhibiting phosphoinositide-3-kinase (PI3K) [53], glucocorticoid receptor activation [56], and JNK and NF-κB pathways [50,57]. Here, 18ga partly attenuated NF-κB translocation (Figure 4), probably via IκBα inhibition, which needs to be confirmed by more experiments. Besides, Ge and Ue partly blocked NF-κB translocation, while other substances did not, which more or less agrees with the NO inhibition results. As p38 mitogen-activated protein kinase (MAPK) was suggested to play a crucial role in regulating LPS-induced NF-κB and activator protein-1 (AP-1) activation as well as iNOS and COX-2 protein expressions [58], it will be interesting to elucidate concrete mechanisms underlining our results. Therefore, further studies are needed.

From a pharmacodynamic point of view, all the six PSMs showed low bioavailability [59,60,61,62,63,64,65]. However, various methods can be used to improve their bioavailability, for example, rectal or nasal administration, or co-administration with fatty acids and sodium deoxycholate/phospholipid-mixed nanomicelles of glycyrrhizic acid [59,60]. Ursolic acid nanoparticles and phospholipid complex could also enhance their bioavailability [66,67,68]. Besides, the phospholipid complex of ursolic acid also exhibited hepatoprotective effect enhancement [68], and a proniosomal gel of ursolic acid showed excellent anti-inflammatory activity [69]. Glycyrrhetinic acid-related nanoparticles were developed to deliver the drug [70,71]. A phosphate prodrug of isoliquiritigenin was developed and showed good stability and solubility [72]; co-crystals of isoliquiritigenin enhanced bioavailability and pharmacokinetic performance [73]; isoliquiritigenin-loaded F127/P123 polymeric micelles showed a strengthened effect [74]. With the help of biotechnological methods, these substances have some potential to be developed as effective drugs.

## 5. Conclusions

Extracts of *Glycyrrhiza glabra* and *Eriobotrya japonica*, 18β glycyrrhetinic acid, and isoliquiritigenin at low concentrations inhibited LPS-induced nitric oxide release in RAW 264.7 cells. 18β glycyrrhetinic acid, isoliquiritigenin, and ursolic acid inhibited the expression of ICAM-1, TNF-α, COX-2, and iNOS, partly through the inhibition of NF-κB expression and the attenuation of NF-κB nuclear translocation. Extracts of *Glycyrrhiza glabra* and *Eriobotrya japonica* also inhibited NF-κB nuclear translocation. These results support the use of *Glycyrrhiza glabra* and *Eriobotrya japonica* and their PSMs in TCM and phytotherapy to treat inflammations.

## Figures and Tables

**Figure 1 medicines-06-00055-f001:**
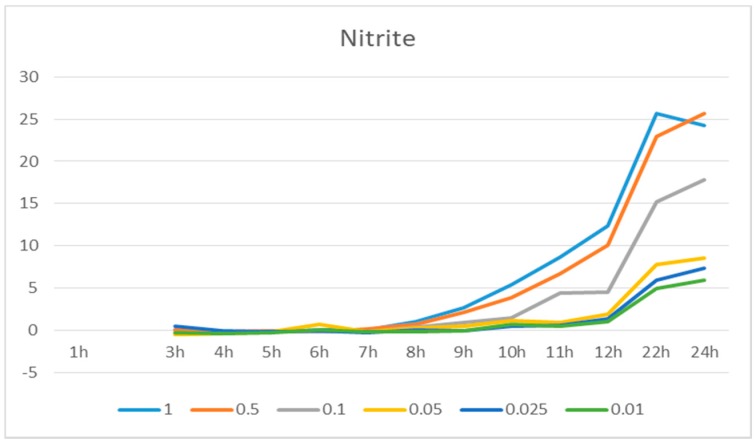
The nitrite concentration (µM) tested at different times with stimulation of different lipopolysaccharide (LPS) doses. Different colors correspond to the different concentrations of LPS (from 0.01 μg/mL to 1 μg/mL).

**Figure 2 medicines-06-00055-f002:**
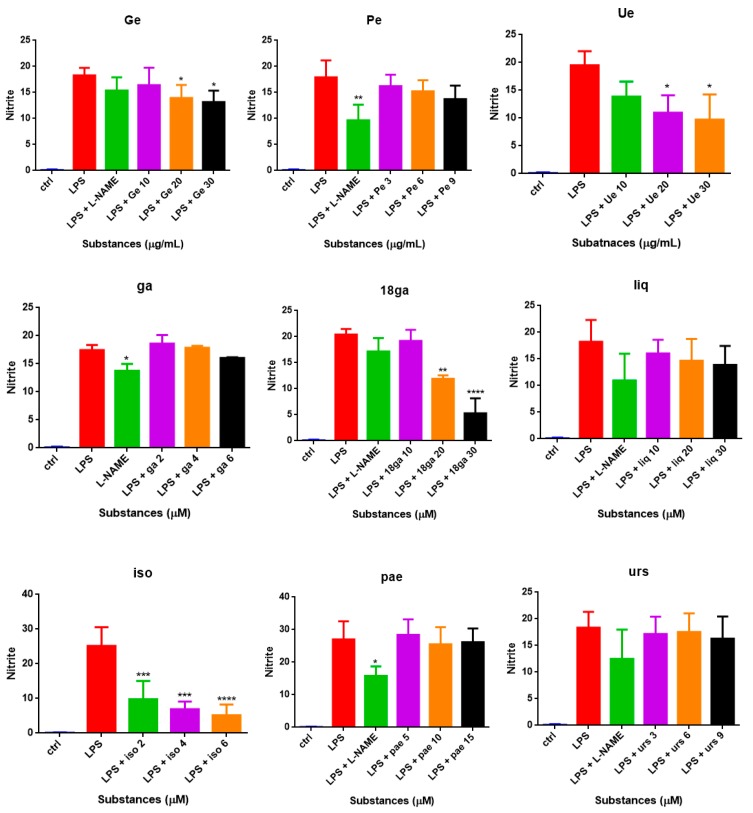
Nitrite levels (µM) in the supernatant of RAW 264.7 cells after treatment with LPS and LPS plus PSMs. The non-treated group was defined as the control group. The results are expressed as the mean ± SD from at least three independent experiments. When *p* < 0.05, the difference was regarded as significant (* *p* < 0.05; ** *p* < 0.01; *** *p* < 0.001; **** *p* < 0.0001).

**Figure 3 medicines-06-00055-f003:**
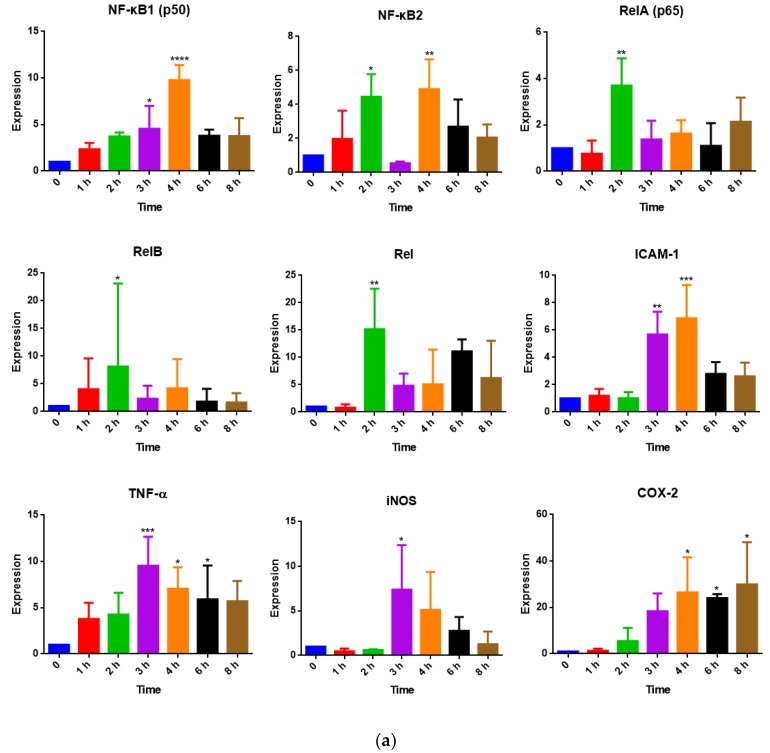
LPS induced NF-κB subunit and NF-κB-regulated gene expression changes. (**a**) NF-κB subunit (NF-κB1, NF-κB2, RelA, RelB, and Rel) and NF-κB-regulated gene (ICAM-1, TNF-α, iNOS, and COX-2) expression changes after 1 μg/mL LPS stimulation for different times. (**b**) Effect of dex, iso, 18ga, or urs on NF-κB subunit and NF-κB-regulated gene expressions at the appointed times. The non-treated group was set as the control group (1.00). The results are expressed as the mean ± SD from at least three independent experiments. When *p* < 0.05, the difference was regarded as significant (* *p* < 0.05; ** *p* < 0.01; *** *p* < 0.001; **** *p* < 0.0001).

**Figure 4 medicines-06-00055-f004:**
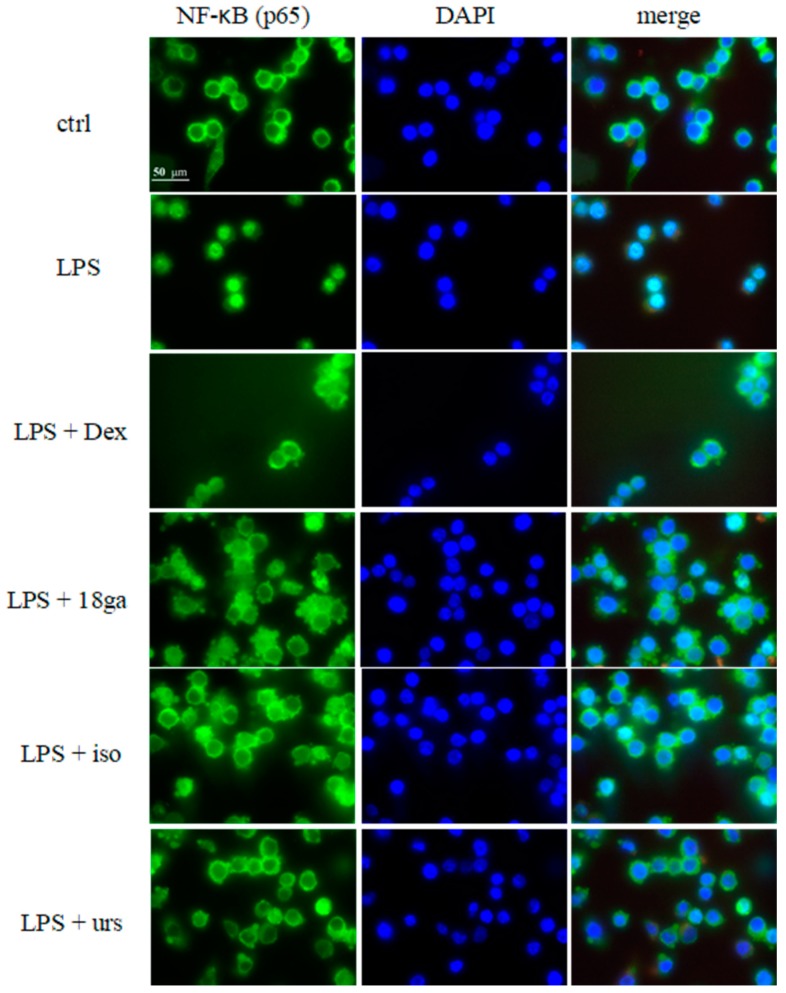
Effect of PSMs on the LPS-induced nuclear translocation of NF-κB. RAW 264.7 cells were treated with LPS or LPS plus plant extracts and PSMs for 2 h. Immunofluorescence and fluorescence microscopy were used to analyze NF-κB translocation. In the first column, the green color was due to the immunofluorescence of an NF-κB secondary antibody; the middle column of blue color shows DAPI-dyed nuclei; and the last column was a merge of the first two columns. When NF-κB moved to the nucleus, the merged nuclei showed a bright white color. The results are expressed as the mean ± SD from at least three independent experiments. (The plotting scale is showed in the control group with p65 staining).

**Table 1 medicines-06-00055-t001:** qPCR primers for NF-κB-related genes.

Gene	Sequence	Amplicon
RPS201 *	Forward: CCCGAAGTGGCGATTCACC	77 nt
	Reverse: TCCGCACAAACCTTCTCCAG	
NF-κB1	Forward: AAGGCAAAGCGAATCCAAA	72 nt
	Reverse: GAAGGCCTTGAATGAAATCG	
NF-κB2	Forward: GTAATCACTGGGCAGACAAGG	95 nt
	Reverse: AAGTGGAGGGCGGAGTCT	
RelA	Forward: TCACCAAGGATCCACCTCA	87 nt
	Reverse: GGCAGAGGTCAGCCTCATAG	
RelB	Forward: TGTTCAAAACGCCACCCTAC	70 nt
	Reverse: CGCTGCAAGAACACATTGAC	
c-Rel #	Forward: AGACTGCGACCTCAATGTGG	117 nt
	Reverse: GCACGGTTGTCATAAATTGGGTT	
ICAM-1	Forward: TGGAAGCTGTTTGAGCTGAG	67 nt
	Reverse: TGCCACAGTTCTCAAAGCAC	
TNF-α	Forward: TTGTCTTAATAACGCTGATTTGGT	61 nt
	Reverse: GGGAGCAGAGGTTCAGTGAT	
iNOS	Forward: CAGACTGGATTTGGCTGGTC	68 nt
	Reverse: CAACATCTCCTGGTGGAACA	
COX-2 +	Forward: TGAGCAACTATTCCAAACCAGC	74 nt
	Reverse: GCACGTAGTCTTCGATCACTATC	

* Primer RPS201 was the reference gene, and its sequences were from PrimerBank MGH-PGA with ID: 146135021c1. https://pga.mgh.harvard.edu/cgi-bin/primerbank/new_displayDetail2.cgi?primerID=146135021c1. # Primer c-Rel sequences were from PrimerBank MGH-PGA with ID: 112181203c3. https://pga.mgh.harvard.edu/cgi-bin/primerbank/new_displayDetail2.cgi?primerID=112181203c3. + Primer COX-2 sequences were from PrimerBank MGH-PGA with ID: 31981525a1. https://pga.mgh.harvard.edu/cgi-bin/primerbank/new_displayDetail2.cgi?primerID=31981525a1. All the primers were synthesized by Eurofins Genomics (Ebersberg, Germany).

**Table 2 medicines-06-00055-t002:** Cytotoxicity (IC_50_ values *) of individual plant extracts and PSMs in RAW 264.7 cells.

Substances	RAW 264.7	Substances	RAW 264.7
Dox ^#^	0.26 ± 0.05	18ga	90.19 ± 2.05
Ge ^#^	94.11 ± 4.88	liq	262.84 ± 26.88
Pe ^#^	67.08 ± 5.31	iso	42.56 ± 5.70
Ue ^#^	60.53 ± 4.03	pae	1034.91 ± 883.85
ga	163.95 ± 74.98	urs	23.40 ± 1.42

* Units for Ge, Pe, and Ue are μg/mL; others are μM; Dox = doxorubicin; Ge = *G. glabra* extract; Pe = *P. lactiflora* extract; Ue = *E. japonica* extract; ga = glycyrrhizic acid; 18ga = 18β glycyrrhetinic acid; liq = liquiritigenin; iso = isoliquiritigenin; pae = paeoniflorin; urs = ursolic acid. ^#^ Values for Dox, Ge, Pe, and Ue were previously stated [38].

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
