# Peer review of "Evidence for Anti-Inflammatory Activity of Isoliquiritigenin, 18β Glycyrrhetinic Acid, Ursolic Acid, and the Traditional Chinese Medicine Plants Glycyrrhiza glabra and Eriobotrya japonica, at the Molecular Level"

_medicines, 2019, doi:10.3390/medicines6020055_

Round 1

Reviewer 1 Report

Dear Authors,

After the review process, I have several comments: 

- you should clarify the aim of the paper (Line 79-84); 

- you should expand section 2.6 (e.g., https://doi.org/10.3390/pharmaceutics11040191); 

- you should comment the bioavailability of the major metabolites; 

- you should detail the possible valorisation in pharmaceutical industry.

Best regards!

Author Response

Dear Reviewer,

Thank you. I tried my best. 

Reviewer 2 Report

The authors continued in their interesting investigations of Paeonia and the other selected plant species of traditional medicine. The authors already published selected results of their research in journal Medicines. 

The used methods were adequate. The data supports the results, which are properly discussed. I appreciate the powerful combination of the used methods. This combination showed to be rational for the anti-inflammatory studies. Since I found no serious errors or mistakes, I recommend the manuscript for publication. 

Author Response

Dear Reviewer, Thank you!

Reviewer 3 Report

The manuscript “Evidence for anti-inflammatory activity of isoliquiritigenin, 18β glycyrrhetinic acid, and ursolic acid and the TCM plants Glycyrrhiza glabra, Paeonia lactiflora, and Eriobotrya japonica at the molecular level” is devoted to the actual problem of natural products. The reviewed article is interesting and theme of the article meets the scope of the journal. However, these anti-inflammatory tests were just several basic in vitro assays, which were not deep and complete enough to prove their effects and to build a manuscript to publish in Medicines. I suggest the authors add at least an in vivo model (paw or ear edema model) to further screen the anti-inflammatory activities of them and resubmit the manuscript.

Author Response

Dear Reviewer,

If time allows, I would be glad to perform an in vivo experiment. 

However, I graduted in March and during the short revision time, it would be difficult. Later if I have a chance, will make up for it. Thank you for your comments.

Round 2

Reviewer 3 Report

Accept in present form.